# Influence of atmospheric conditions on the role of trifluoroacetic acid in atmospheric sulfuric acid-dimethylamine nucleation

Ling Liu[1], Fangqun Yu[2], Kaipeng Tu[1], Zhi Yang[1], and Xiuhui Zhang[1]

[1]Key Laboratory of Cluster Science, Ministry of Education of China, School of Chemistry and Chemical Engineering, Beijing Institute of Technology, Beijing 100081, China
[2]Atmospheric Sciences Research Center, University at Albany, Albany, New York 12203, USA

**Correspondence:** Xiuhui Zhang (zhangxiuhui@bit.edu.cn)

**Abstract.** Ambient measurements combined with theoretical simulations have shown evidence that the tropospheric degradation end-products of Freon alternatives, trifluoroacetic acid (TFA), one of the most important and abundant atmospheric organic substances, can enhance the process of sulfuric acid (SA) - dimethylamine (DMA) - based nucleation process in urban environments. However, TFA is widespread all over the world with different atmospheric conditions, such as temperature and nucleation precursor concentration, which are the most important factors potentially influencing the atmospheric nucleation process and thus inducing different nucleation mechanisms. Herein, using the Density Functional Theory combined with the Atmospheric Cluster Dynamics Code, the influence of temperature and nucleation precursor concentrations on the role of TFA in the SA-DMA nucleation has been investigated. The results indicate that the growth trends of clusters involving TFA can increase with the decrease of temperature. The enhancement on particle formation rate by TFA and the contributions of the SA-DMA-TFA cluster to the cluster formation pathways can be up to 227 times and 95%, respectively, at relatively low temperature, low SA concentration, high TFA concentration, and high DMA concentration, such as in winter, at the relatively high atmospheric boundary layer, or in megacities far away from industrial sources of sulfur-containing pollutants. These results provide the perspective of the realistic role of TFA in different atmospheric environments, revealing the potential influence of the tropospheric degradation of Freon alternatives under a wide range of atmospheric conditions.

## 1 Introduction

Atmospheric aerosols have significant adverse impacts on climate and human health (Kulmala et al., 2007; Stevens and Feingold, 2009). New particle formation (NPF) is not only the key process of the formation of atmospheric aerosol but also the source of over half of the atmospheric cloud condensation nuclei, thus influencing cloud properties and Earth's energy balance (Bianchi et al., 2016; Merikanto et al., 2009). Especially, nucleation is the initial critical step of NPF. Well understanding the nucleation mechanisms and compositions of nucleation precursors can help to predict the impacts of NPF events and further provide theoretical clues to reducing the severe atmospheric aerosol pollution, haze. Sulfuric acid has been well recognized as the key nucleation precursor (Doyle, 1961; Kulmala et al., 1995; Berndt et al., 2005; Kulmala et al., 2013). Although sulfuric acid (SA) - dimethylamine (DMA) nucleation mechanism has been observed in various places around the world (Yao et al., 2018; Deng et al., 2020; Brean et al., 2020), there are still a lot of species observed in the atmosphere but not fully assigned

molecular formulas because of their chemical complexity. Previous studies have identified that some acids, such as methane sulfonic acid, sulfamic acid, and glyoxylic acid, can enhance the SA-based particle formation (Bork et al., 2014; Li et al., 2018; Liu et al., 2017). However, other species that can potentially enhance the NPF rates and the corresponding nucleation mechanism still should be further explored.

Perfluorocarboxylic acids (PFCAs), widely distributed in the environment, can be formed from the oxidation of anthropogenically produced hydrofluorocarbons (HFCs), hydrochlorofluorocarbons (HCFCs), and hydrofluoro-olefins (HFOs) (Kazil et al., 2014; Burkholder et al., 2015; Pickard et al., 2018). Due to their worldwide distribution, chemical stability, and potential biological toxicity, PFCAs are generally believed to be an important class of environmental contaminants present in various environments (Wang et al., 2020; Cheng et al., 2020). As the simplest and most abundant perfluorocarboxylic acid in the atmosphere (Tian et al., 2018), trifluoroacetic acid (TFA) is the pivotal product of the oxidative degradation processes of the Freon alternatives emitted from human activities (Madronich et al., 2015; Tromp et al., 1995). The concentration of TFA is relatively high in regions with scant rainfall because precipitation is the only predominant environmental sink of TFA (Ellis et al., 2001). Thus, the continuous production of TFA from the degradation of Freon alternatives combined with the lack of rainfall in some areas is more likely to induce the accumulation of TFA. Recent ambient measurements and theoretical simulations have shown evidence for the participation of TFA in the formation of SA-DMA-based clusters under certain atmospheric conditions in urban Shanghai, China (Lu et al., 2020). The participation of TFA in NPF events can increase the number concentration of atmospheric aerosol particles and further potentially have effects on climate and human health.

The previous study on the role of TFA in the SA-DMA-based NPF process is under the local atmospheric temperature and nucleation precursor concentrations of Shanghai (Lu et al., 2020). However, TFA is widely distributed all over the world with different atmospheric conditions, among which temperature and concentrations of nucleation precursors are key influential factors in the nucleation process. Not only can temperature change with the variations of seasons, altitudes, and climates, but also nucleation precursor concentrations can vary with altitudes and distances to corresponding sources. For example, DMA concentration is relatively high in polluted regions near its industrial, residential, or agricultural sources, but it is relatively low in the clean and upper troposphere due to its short lifetime in the atmosphere. SA concentration is relatively high in areas near coal-fired power plants or industries, but it is relatively low in areas far away from the emission sources of sulfur-containing pollutants. Besides, although TFA is widely distributed around the world, its concentration is relatively high in regions with scant rainfall. Therefore, it is important to elucidate the role of TFA in the NPF events under broad atmospheric conditions to understand the effect of TFA on the atmospheric environment systematically. In the present study, the influence of the varying temperatures and precursor concentrations on the role of TFA in the SA-DMA-based clustering process was studied using Density Functional Theory combined with the Atmospheric Cluster Dynamics Code (ACDC) (McGrath et al., 2012; Olenius et al., 2013). The studied clusters are $(\text{Acid})_m \cdot (\text{Base})_n$ $(0 \leq n \leq m \leq 3)$, in which acid molecules are TFA or/and SA and base molecule is DMA.

## 2 Methods

The M06-2X functional has been successfully used to describe noncovalent interaction (Elm et al., 2012; Zhao and Truhlar, 2008) and estimate the thermochemistry, equilibrium structures of atmospheric clusters (Bork et al., 2014; Elm et al., 2012, 2013). Furthermore, the 6-311++G(3df,3pd) basis set was chosen based on its excellent performance to estimate the properties of atmospheric relevant clusters when used in conjunction with the M06-2X functional (Herb et al., 2011, 2013; Nadykto et al., 2011, 2009). The single-point electronic energies were corrected using the RI-CC2 method and the aug-cc-pV(T+d)Z basis set performed with the TURBOMOLE program, because of the good agreement between the simulated results based on the single-point electronic energies at the RI-CC2/aug-cc-pV(T+d)Z level of theory and the experimental or field measurements (Kürten et al., 2018; Lu et al., 2020). Structures and the thermodynamic data of the presently studied clusters at 280 K were taken from previous studies (Lu et al., 2020).

The Atmospheric Cluster Dynamics Code (ACDC) (McGrath et al., 2012; Olenius et al., 2013) was used to simulate the cluster formation process with the thermodynamic data generated by quantum chemistry calculations as input. Time development of the concentration of each cluster was solved by integrating the birth-death equation numerically (Kulmala et al., 2001) using the ode15s solver in MATLAB-R2013a program (Shampine and Reichelt, 1997).

The birth-death equation can be written as following Eq. (1):

$$\frac{dc_i}{dt} = \frac{1}{2}\Sigma_{j<i}\beta_{j,(i-j)}c_j c_{(i-j)} + \Sigma_j \gamma_{(i+j)\to i}c_{i+j} - \Sigma_j \beta_{i,j}c_i c_j - \frac{1}{2}\Sigma_{j<i}\gamma_{i\to j}c_i + Q_i - S_i \tag{1}$$

where $c_i$ is the concentration of cluster $i$, $\beta_{i,j}$ is the collision coefficient between clusters $i$ and $j$, $\gamma_{i\to j}$ is the evaporation coefficient of a molecule or a smaller cluster $j$ from cluster $i$, $Q_i$ is an outside source term of cluster $i$, and $S_i$ is other possible sink term of cluster $i$.

The collision rate coefficients, $\beta_{i,j}$, between clusters $i$ and $j$ were calculated as hard-sphere collision (Chapman et al., 1990) in Eq. (2):

$$\beta_{i,j} = \pi \left(r_i + r_j\right)^2 \sqrt{\frac{8k_B T}{\pi\mu}} \tag{2}$$

where $r_i$ is the radius of cluster $i$ given by Multiwfn 3.3.8 program (Lu and Chen, 2012), $k_B$ is the Boltzmann constant, $T$ is the temperature, and $\mu = m_i m_j / (m_i + m_j)$ is the reduced mass. The cluster radius is half of the sum of the distance between the center of most distant atoms in the cluster given by the Multiwfn 3.3.8 program and the Van der Walls radii of these atoms.

Evaporation coefficients, $\gamma_{i\to j}$, were obtained from the corresponding collision coefficients and $\Delta G$ of clusters as shown in Eq. (3):

$$\gamma_{(i+j)\to i} = \beta_{i,j}\frac{P_{ref}}{k_B T}\exp\left(\frac{\Delta G_{i+j} - \Delta G_i - \Delta G_j}{k_B T}\right) \tag{3}$$

where $P_{ref}$ is the reference pressure (in this case 1 atm), at which the formation free energies were calculated and $\Delta G_i$ is the Gibbs free energy of formation of cluster $i$ from monomers.

## 3 Results and Discussion

### 3.1 Influence of atmospheric conditions on the stability and growth trend of clusters

The cluster stability is of great significance for the process of NPF. Atmospheric temperatures can vary with different seasons, altitudes, and climates. The temperature variation may have a significant influence on the aerosol cluster stability. Thermodynamically, Gibbs free energies of formation ($\Delta G$) for clusters at different temperatures (280 K and 260 K) within the temperature range of the atmospheric boundary layer are shown in Table S1 in the Supplement. It shows that $\Delta G$ decreases with the decrease of temperature, which reflects that clusters are more thermodynamically stable at relatively lower temperatures.

Kinetically, if the collision frequency ($\beta \cdot C$, $\beta$ is shown in Table S2 in the Supplement) of a cluster with a monomer molecule at the concentration of $C$ is higher than its total evaporation frequency ($\Sigma\gamma$, Table S3 in the Supplement), this cluster can have the potential to continue growing. Thus, in order to analyze the growth trends of clusters involving TFA, ratios ($\beta \cdot C/\Sigma\gamma$) for collision frequencies with nucleation monomers versus total evaporation frequencies have been investigated (Table S4 in the Supplement). Here, we take the ratio of $\beta \cdot C/\Sigma\gamma$ for unhydrated clusters involving one TFA molecule at different temperatures shown in Fig. 1 as an example to study the growth trend of clusters. The $\beta \cdot C/\Sigma\gamma$ increases with the decrease of temperature (Fig. 1), which indicates that it is easier for these clusters to grow at lower temperatures. The reason for this is that the influence of temperature variation on the evaporation coefficients ($\gamma$, where the temperature is in the exponential term (Eq. (3))) is much greater than that on collision coefficients ($\beta$, where the temperature is in the square root term (Eq. (2))) (McGrath et al., 2012). At relatively low concentration of SA monomer, the ratios of $\beta \cdot C/\Sigma\gamma$ for $(SA)_1 \cdot (DMA)_2 \cdot (TFA)_1$ and $(SA)_2 \cdot (DMA)_3 \cdot (TFA)_1$ clusters are all larger than 1.0 at different temperatures (Fig. 1), which means that these two kinds of clusters tend to grow to larger clusters at 280 K and 260 K. Besides, $\beta \cdot C/\Sigma\gamma$ will increase when the nucleation monomer concentration increases. Thus, the growth trend of clusters can also be relatively high at higher concentrations of nucleation monomers. In addition, in order to evaluate the influence of relative humidity on the formation kinetics of the SA-DMA-TFA system, the hydrated key clusters have also been studied. The relative hydrate distributions, the ratios of effective collision coefficients to evaporation coefficients weighted average over the hydrate distributions (Paasonen et al., 2012), and Cartesian coordinates of hydrated clusters are listed in Tables S5-S6, Table S7, and Table S8 in the Supplement, respectively. From the relevant detailed discussion shown in Section S1 in the Supplement, we can see that the stability of the studied SA-DMA-TFA clusters is insensitive to relative humidity, which is similar to the SA-DMA clusters (Olenius et al., 2017). Thus, only the unhydrated SA-DMA-TFA clusters are considered in the following discussion. The boundary of the ACDC simulation should be set as the smallest clusters that are stable enough to grow outside of the simulated system (McGrath et al., 2012). Based on the stability of the studied clusters, $(SA)_4 \cdot (DMA)_3$ and $(SA)_3 \cdot (DMA)_4 \cdot (TFA)_1$ clusters are set to be the boundary, which is consistent with that in the previous study (Lu et al., 2020).

### 3.2 Influence of temperature and nucleation precursor concentrations on particle formation rates

It can be seen that the stability and growth trend of clusters involving TFA vary with temperatures and concentrations of nucleation precursors from the above discussion. The stability and growth trend are closely related to particle formation rates. A

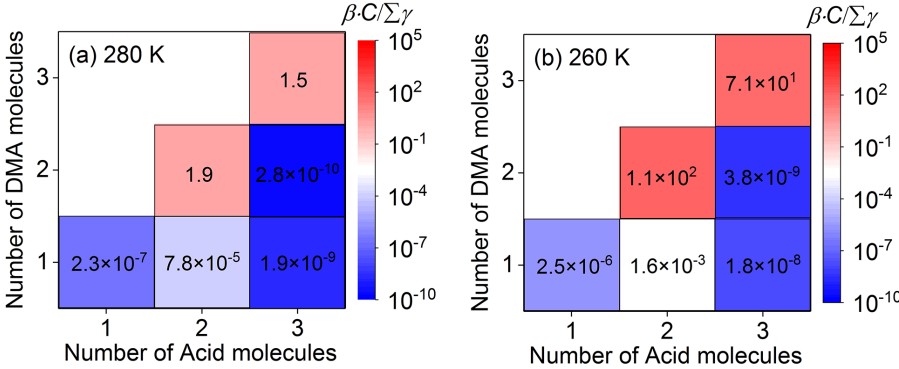

**Figure 1.** Ratios of SA monomer collision frequencies versus total evaporation frequencies ($\beta \cdot C / \Sigma \gamma$) for $(SA)_x \cdot (DMA)_y \cdot (TFA)_1$ ($0 \leq x \leq 2$, $1 \leq y \leq 3$, $y \leq x+1$) clusters at different temperatures (280 K and 260 K). Acid molecules are SA and TFA. $C = 1.0 \times 10^6$ molecules cm$^{-3}$.

study on the complicated influence of temperature and nucleation precursor concentration on particle formation rates becomes increasingly important to understand the realistic role of TFA in different atmospheric environments. Therefore, the potential influences of temperatures and concentrations of nucleation precursors on the particle formation rates of the SA-DMA-TFA system have been investigated below.

The temperature was obtained from the NASA Langley Research Center (LaRC) POWER Project funded through the NASA Earth Science/Applied Science Program (NASA, 2020). The studied temperature is the monthly average of temperature at 2 meters above the surface of the earth for a given month, averaged for that month over the 30 years (Jan. 1984 – Dec. 2013). The lifetime of TFA and SA is long enough to keep their atmospheric concentrations within the studied range. However, because of the relatively short atmospheric lifetime of DMA concerning chemical reaction loss (Carl and Crowley, 1998; Qiu and Zhang, 2013), whether the concentration of DMA ([DMA]) can be relatively high or not remains to be determined. The global chemistry-transport model of GEOS-Chem (Yu and Luo, 2014) can predict the global [DMA], but the [DMA] may be underestimated possibly because of the underestimation about emissions and the regional average (Yu and Luo, 2014). Therefore, the cluster formation rates and the enhancements on cluster formation rate by TFA have been simulated at five times the predicted [DMA] by the global chemistry-transport model of GEOS-Chem (Yu and Luo, 2014), taking four different typical cities with relatively high [DMA], such as Beijing, Shanghai, Los Angeles, and New Delhi, as examples (Fig. 2). The corresponding temperature and [DMA] in different months of these four cities are listed in Table S9, and the $\Delta G$ at the relevant temperatures is listed in Table S10 in the Supplement. The SA concentration ([SA]) and TFA concentration ([TFA]) were chosen to be relatively low and high, $5.0 \times 10^6$ molecules cm$^{-3}$ and $1.0 \times 10^8$ molecules cm$^{-3}$ (Wu et al., 2014) respectively. The condensation sink (CS) is set to be 0.02 s$^{-1}$, which is the median of common CSs in the NPF events in polluted areas (Yao et al., 2018).

As shown in Fig. 2, the enhancements on particle formation rate by TFA in the studied four cities are all higher than 1.0 in most of the months throughout the year, which indicates that TFA can enhance the SA-DMA particle formation rate in

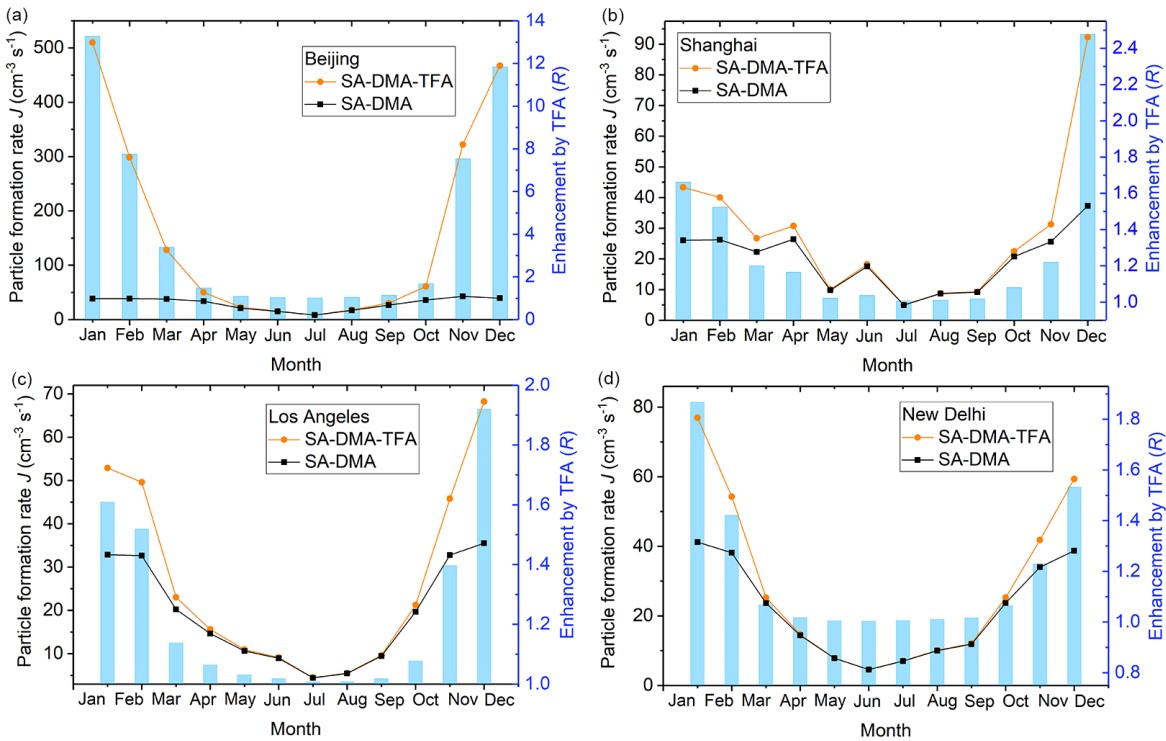

**Figure 2.** Particle formation rates ($J$, cm$^{-3}$ s$^{-1}$) shown in orange circles (SA-DMA-TFA) and black squares (SA-DMA) and the enhancement on particle formation rates by TFA ($R$, $R = J_{\mathrm{SA-DMA-TFA}}/J_{\mathrm{SA-DMA}}$) shown in bar charts in (a) Beijing, (b) Shanghai, (c) Los Angeles, and (d) New Delhi.

most parts of the year. The particle formation rate and the enhancement on particle formation rate by TFA in Beijing both are almost higher than those in the other three studied cities in the corresponding month. This can be attributed to the fact that the [DMA] and temperature in Beijing are relatively higher and lower than those in the other three studied cities, respectively.

Besides, the particle formation rate and enhancement on particle formation rate by TFA are relatively high in spring and winter approximatively. This can be attributed to the fact that the temperature in spring and winter is relatively lower than that in other seasons all the year round. Moreover, the enhancement on the particle formation rate by TFA is highest in Beijing among the studied four cities, which can be up to more than 13 times in January at [DMA] of $4.69 \times 10^8$ molecules cm$^{-3}$ and 265 K. Therefore, there is enough DMA in the realistic atmospheric boundary layer for TFA to enhance the SA-DMA nucleation

process.

The influences of different atmospheric conditions on the effects of TFA in SA-DMA nucleation have been further studied in a wide range of common concentrations of nucleation precursors at common temperatures (280 K and 260 K) in the atmospheric boundary layer (Fig. 3). The condensation sink (CS) in Fig. 3 is set to be 0.02 s$^{-1}$, which is the median of common CSs in the NPF events in the polluted areas (Yao et al., 2018). The results at other common CSs (0.01 s$^{-1}$ and 0.03

s$^{-1}$) in the polluted atmosphere will be discussed below. DMA is one of the most common basic nucleation precursors in the

atmosphere and can be released from industrial, residential, or agricultural sources (Mao et al., 2018). The common [DMA] in the atmosphere is in the range of $1.0 \times 10^7$ to $1.0 \times 10^9$ molecules cm$^{-3}$. In this DMA concentration range, the particle formation rate and the enhancement of TFA on particle formation rate ($R$, $J_{\text{SA－DMA－TFA}}/J_{\text{SA－DMA}}$) have been studied at moderate atmospheric [SA] and [TFA]. In general, the particle formation rate of the SA-DMA-TFA system (Fig. 3 (a)) and the enhancement on particle formation rate by TFA (Fig. 3 (b)) increase with the decrease of temperature and increase with the increase of [DMA], which is in accordance with the negative dependence of the growth trend of clusters involving TFA on temperature and positive dependence on nucleation monomer concentration. At [DMA] = $1.0 \times 10^9$ molecules cm$^{-3}$ and 260 K, the particle formation rate of the SA-DMA-TFA system can be up to more than 1200 cm$^{-3}$ s$^{-1}$, and the corresponding enhancement on particle formation rate by TFA can nearly reach up to 2.2 times. Thus, the enhancement on particle formation rate by TFA is obvious in the regions with relatively low temperature and abundant DMA, such as in winter or at the relatively high atmospheric boundary layer of industrial regions, megacities with a high population density, or rural areas with animal housing and grazing regions (Mao et al., 2018).

In addition to the basic molecules, SA is the key acidic nucleation precursor in the atmosphere, and its main sources are the coal-fired power plants and the industries. The common atmospheric [SA] is in the range of $1.0 \times 10^6$ to $1.0 \times 10^8$ molecules cm$^{-3}$ from clean areas to highly polluted areas (Kürten et al., 2012; Yao et al., 2018; Zheng et al., 2015). Given that the enhancement of TFA on particle formation rate is high at relatively high [DMA], the influence of [SA] on the role of TFA in NPF at [DMA] = $1.0 \times 10^9$ molecules cm$^{-3}$ and different temperatures (280 K and 260 K) has been further studied. At different [SA], there are negative dependencies of the particle formation rate and the enhancement on formation rate by TFA on temperature. At different temperatures, the particle formation rates all increase with the increase of [SA] (Fig. 3 (c)). However, the enhancement of particle formation rate by TFA increases with the decrease of [SA] and can be more than 14 times at relatively low [SA] ($1.0 \times 10^6$ molecules cm$^{-3}$) at 260 K (Fig. 3 (d)). Thus, in addition to the relatively high [DMA], the enhancement on particle formation rate by TFA is obvious in the regions with relatively low temperature and low [SA], such as in winter or at the relatively high atmospheric boundary layer of megacities and rural regions both far away from industrial sources of sulfur-containing pollutants, such as Beijing, China. Especially, with the effective regulations of emission reduction of the sulfur-containing pollutants, the atmospheric [SA] will decrease and the role of TFA (one of other important and potential acidic nucleation precursors besides SA) in NPF will become increasingly important.

Furthermore, TFA is widely distributed in the atmosphere, especially in areas with scant rainfall. The variation of [TFA] may directly influence its role in the process of NPF. At relatively high [DMA] of $1.0 \times 10^9$ molecules cm$^{-3}$ and relatively low [SA] of $1.0 \times 10^6$ molecules cm$^{-3}$, the influence of [TFA] on particle formation rate has been further studied at different temperatures (280 K and 260 K). Similar to the influence of temperature at different [DMA] or [SA], there are negative dependencies of the particle formation rate and the enhancement on particle formation rate by TFA on the temperature at different [TFA]. The particle formation rate (Fig. 3 (e)) and the enhancement of TFA on particle formation rate increase (Fig. 3 (f)) with the increase of [TFA] at different temperatures, which is in accordance with the positive dependence of the growth trend for clusters on nucleation monomer concentration. At relatively high [TFA] ($1.0 \times 10^8$ molecules cm$^{-3}$) and at a relatively low temperature of 260 K, the particle formation rate is more than 17 cm$^{-3}$ s$^{-1}$, and the corresponding enhancement on particle formation

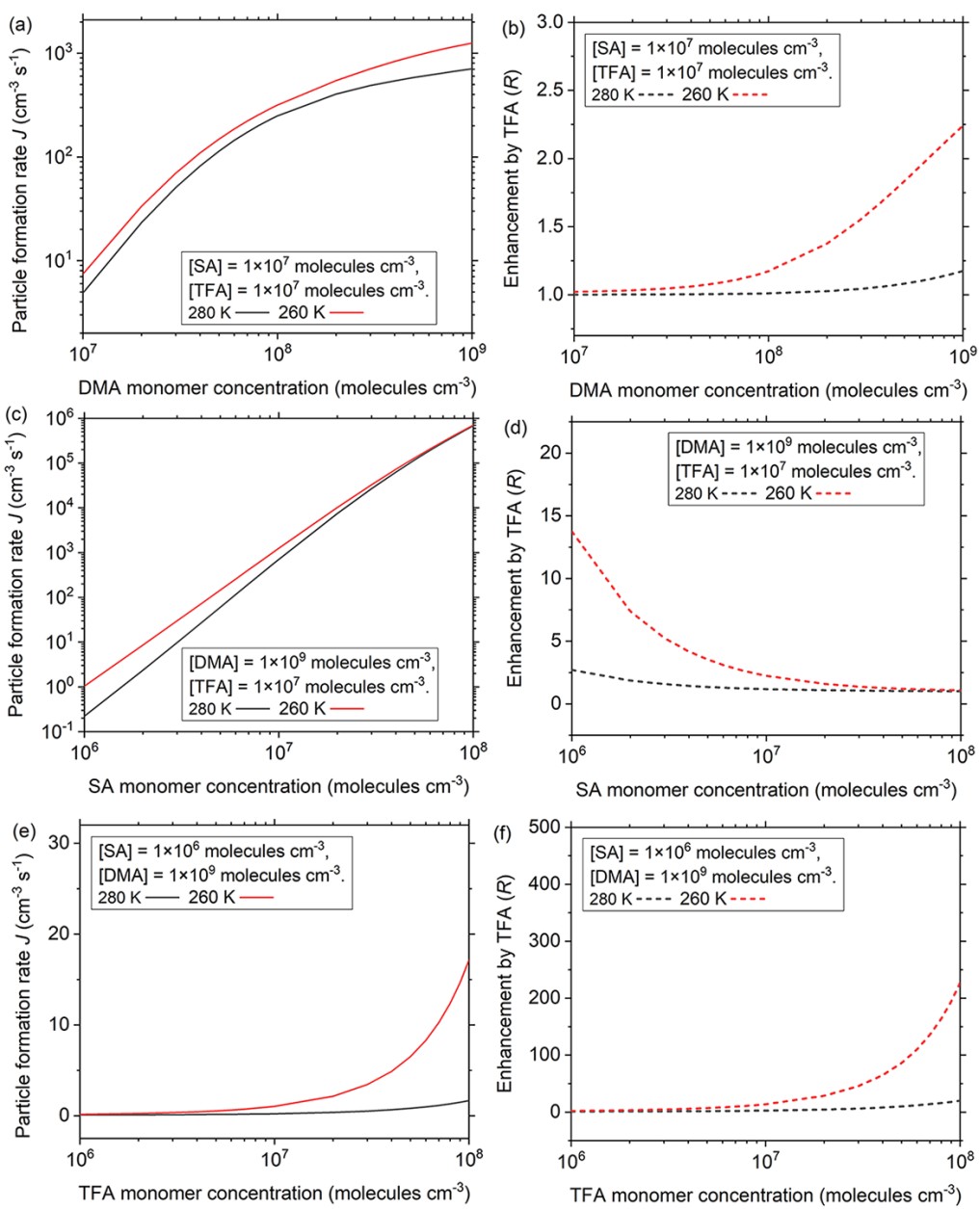

**Figure 3.** Particle formation rates ($J$, cm$^{-3}$ s$^{-1}$) at different temperatures (280 K and 260 K) as a function of (a) DMA monomer concentrations, (c) SA monomer concentrations, and (e) TFA monomer concentrations. Enhancement on particle formation rate by TFA ($R$, $R =$ $J_{\mathrm{SA-DMA-TFA}}/J_{\mathrm{SA-DMA}}$) at different temperatures as a function of (b) DMA monomer concentrations, (d) SA monomer concentrations, and (f) TFA monomer concentrations. Black and red lines are corresponding to 280 K and 260 K, respectively. CS = 0.02 s$^{-1}$.

rate by TFA can be as high as more than 227 times. Therefore, the role of TFA is significant in the regions with relatively low temperatures and high [TFA], such as in winter or at the relatively high atmospheric boundary layer and megacities or rural regions with scant rainfall far away from industrial sources of sulfur-containing pollutants.

Relatively high aerosol concentration can cause a large condensation sink (CS), thus scavenging newly formed molecular clusters (Yao et al., 2018). The CS can be different in different atmospheric environments. The particle formation rates and the corresponding enhancements by TFA at other common CSs ($0.01 \text{ s}^{-1}$ and $0.03 \text{ s}^{-1}$) in the polluted atmosphere are respectively shown in Figs. S1 and S2 in the Supplement. Although the particle formation rate and enhancement by TFA increase with the decrease of CS, the influence trend of temperature and nucleation precursor concentrations on them at different CSs are the same.

## 3.3 Influence of temperature and precursor concentrations on the formation pathway of SA-DMA-TFA clusters

Variations of atmospheric conditions (temperature and concentration) can influence the particle formation rate of the SA-DMA-TFA system as well as the corresponding enhancement by TFA. These trends of influence can be potentially explained by the detailed SA-DMA-TFA cluster formation mechanism at the molecular level. Thus, the detailed cluster formation mechanisms of the SA-DMA-TFA system under different temperatures and different concentrations of nucleation precursors have been further explored.

The cluster formation pathway at different concentrations of nucleation precursors (DMA, SA, and TFA) are first studied at the common temperature (280 K) of the atmospheric boundary layer (Fig. 4 (a)). The SA-DMA-TFA cluster formation pathways at different [DMA] have been shown at moderate values of [SA] = $1.0 \times 10^7$ molecules cm$^{-3}$ and [TFA] = $1.0 \times 10^7$ molecules cm$^{-3}$. There are two kinds of cluster formation pathways including the pure SA-DMA cluster formation and the SA-DMA-TFA cluster formation. The clusters involving TFA can be formed initially by the addition of one TFA molecule to $(SA)_1 \cdot (DMA)_1$ cluster, and continue growing by the addition of one DMA molecule and then the addition of one $(SA)_1 \cdot (DMA)_1$ cluster. Other studied clusters involving TFA can't present in the main cluster formation pathway because of their relatively low stability. The contributions of SA-DMA-TFA pathways (Fig. 4 (a-1)) increase with the increase of [DMA], which can explain why there is a positive dependence of the SA-DMA-TFA particle formation rate on [DMA]. When [DMA] increases up to $1.0 \times 10^9$ molecules cm$^{-3}$, the cluster formation pathway involving TFA (blue arrows in Fig. 4 (a)) can contribute to the cluster formation by up to 13% (4% + 9%). Therefore, there is a non-negligible contribution of clusters involving TFA to the particle formation pathway in regions with relatively high [DMA], such as regions near industrial, residential, or agricultural sources.

The SA-DMA-TFA cluster formation pathways at different [SA] have been studied at the relatively high [DMA] ($1.0 \times 10^9$ molecules cm$^{-3}$) in view of that the contribution of TFA to the cluster formation pathway can be relatively high at high [DMA] shown by the above results. The moderate value of [TFA] ($1.0 \times 10^7$ molecules cm$^{-3}$) and common temperature (280 K) of the atmospheric boundary layer were chosen. The contributions of pathways at different [SA] (Fig. 4 (a-2)) are different. As shown in Fig. 4 (a-2), the contributions of branch pathways involving TFA increase with the decrease of [SA], which can be up to 61% (41% + 20%) at [SA] = $1.0 \times 10^6$ molecules cm$^{-3}$. Additionally, the cluster formation pathways with the variations of [TFA]

at the relatively low [SA] ($1.0 \times 10^6$ molecules cm$^{-3}$) and the relatively high [DMA] ($1.0 \times 10^9$ molecules cm$^{-3}$) are also shown in Fig. 4 (a-3). As expected, the contribution of pathways involving TFA to the cluster formation pathway is positively dependent on [TFA]. Especially, this contribution can be up to 93% at the relatively low [SA], high [DMA], and high [TFA] (Fig. 4 (a-3)), which indicates that the cluster formation pathway involving TFA can dominate in megacities with scant rainfall far away from industrial sources of sulfur-containing pollutants. These variation trends of contributions of the SA-DMA-TFA

pathway are in accordance with the above discussion that there are a negative dependence and a positive dependence of the SA-DMA-TFA particle formation rate on [SA] and [TFA], respectively.

  The SA-DMA-TFA cluster formation pathways at relatively low [SA], high [DMA], and high [TFA] are further studied at 260 K (Fig. 4 (b)). Compared to the cluster formation pathway at 280 K, additional pathways shown in blue arrows are present at 260 K. In addition to the two independent pathways of the pure SA-DMA pathway and the SA-DMA-TFA pathway starting

from $(SA)_1 \cdot (DMA)_1$ cluster, $(SA)_2 \cdot (DMA)_2 \cdot (TFA)_1$ cluster can contribute to the formation of $(SA)_2 \cdot (DMA)_2$ cluster by the evaporation of one TFA molecule. The contributions of the cluster formation pathway involving TFA can be up to 95% (18% + 77%). Therefore, the participation of TFA in the cluster formation process is extensive at relatively low temperatures, such as in winter or at the relatively high atmospheric boundary layer, which is in accordance with the negative dependence of SA-DMA-TFA cluster formation rates on temperature.

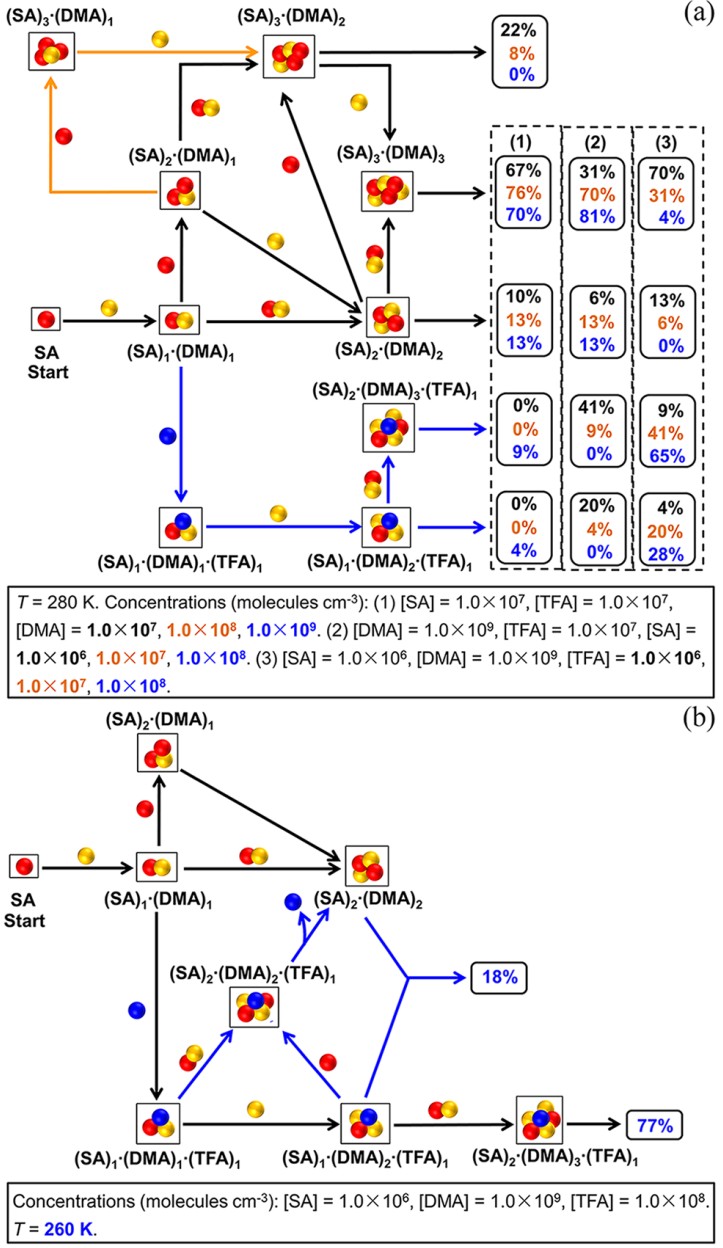

**Figure 4.** (a) Main cluster formation pathways of SA-DMA-TFA clusters at different [DMA], [SA], and [TFA]. The values of contribution shown in the right frames in black, orange, and blue are corresponding to the contributions at (1) [DMA] of $1.0 \times 10^7$ molecules cm$^{-3}$, $1.0 \times 10^8$ molecules cm$^{-3}$ and $1.0 \times 10^9$ molecules cm$^{-3}$, respectively, (2) [SA] of $1.0 \times 10^6$ molecules cm$^{-3}$, $1.0 \times 10^7$ molecules cm$^{-3}$ and $1.0 \times 10^8$ molecules cm$^{-3}$, respectively, and (3) [TFA] of $1.0 \times 10^6$ molecules cm$^{-3}$, $1.0 \times 10^7$ molecules cm$^{-3}$ and $1.0 \times 10^8$ molecules cm$^{-3}$, respectively. Black arrows show the common pathways present at all the studied [DMA]. Orange and blue arrows show the additional pathways present at [DMA] = $1.0 \times 10^7$ molecules cm$^{-3}$ and [DMA] = $1.0 \times 10^9$ molecules cm$^{-3}$. (b) Main cluster formation pathways of SA-DMA-TFA clusters at 260 K. Black arrows show the common pathways present at both 260 K and 280 K. Blue arrows show the additional pathways present at 260 K compared with 280 K.

## 4 Atmospheric implications and conclusions

The present study shows that the influence of atmospheric conditions on the role of TFA in SA-DMA nucleation by the combination of high-level quantum-chemical calculations with the Atmospheric Cluster Dynamics Code (ACDC) simulations in the typical ranges of atmospheric temperature and nucleation precursor concentration. Results show a negative dependence of the growth trend of clusters involving TFA on temperature and a positive dependence on nucleation precursor concentration. The enhancement on particle formation rate by TFA and the contributions of the SA-DMA-TFA cluster formation pathway to the main cluster pathways increase with the decrease of temperature, the increase of [DMA] and [TFA], and the decrease of [SA]. At [DMA] = $1.0 \times 10^9$ molecules cm$^{-3}$, [TFA] = $1.0 \times 10^8$ molecules cm$^{-3}$, [SA] = $1.0 \times 10^6$ molecules cm$^{-3}$ and 260 K, the enhancement on SA-DMA particle formation rate by TFA can be as much as 227 times with the particle formation rate being more than 17 cm$^{-3}$ s$^{-1}$. Under these atmospheric conditions, the corresponding contribution of clusters involving TFA in the cluster formation pathway is about 95% at 260 K. Therefore, the role of TFA in atmospheric aerosol nucleation is of great significance in regions with relatively low temperature, high [DMA], and [TFA], and low [SA], such as cold megacities with scant rainfall. Especially, with the effective regulations of emission reduction of sulfur-containing pollutants, the enhancement of TFA on NPF becomes increasingly important. Under these atmospheric conditions, TFA can extensively participate in the SA-DMA-based cluster formation process and further significantly enhance the particle formation rate, which can potentially enhance the number concentrations of cloud condensation nuclei and further influence the climate. Furthermore, this study inspires the potential influence of the usage of Freon alternatives on air quality and global climate under different atmospheric conditions.

*Data availability.* The data in this article are available from the corresponding author upon request (zhangxiuhui@bit.edu.cn)

*Author contributions.* LL and XHZ designed the research. LL and KKT conducted the quantum chemistry calculations and the simulation by Atmospheric Cluster Dynamic Code. FQY conducted the simulation by global chemistry-transport model of GEOS-Chem. LL analyzed data with the contributions from XHZ, FQY and ZY. LL and XHZ wrote the paper with contributions from all of the other co-authors.

*Competing interests.* The authors declare that they have no conflict of interest.

*Acknowledgements.* This work was supported by the National Natural Science Foundation of China (21976015, 21673175) and the China Postdoctoral Science Foundation (2020M680013). We thank the useful help of Theo Kurtén (University of Helsinki) and Hanna Vehkamäki (University of Helsinki). We acknowledge National Supercomputing Center in Shenzhen for providing the computational resources and Turbomole.

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
