# Peer review of "Influence of atmospheric conditions on the role of trifluoroacetic acid in atmospheric sulfuric acid-dimethylamine nucleation"

_Atmospheric Chemistry and Physics, 2020_

## Referee Comment (RC1) · Anonymous Referee #1 · 4 Jan 2021

Liu and co-workers investigated the influence of different atmospheric conditions on the role of trifluoroacetic acid (TFA) in sulfuric acid (SA) – dimethylamine (DMA) nucleation process using the Density Functional Theory combined with the Atmospheric Cluster Dynamics Code, which is based the previous study on that TFA can participate in SA-DMA-based nucleation under the local atmospheric temperature and nucleation precursor concentrations of Shanghai, China. This study reports the enhancement of particle formation rate by TFA and the contributions of SA-DMA-TFA cluster to the cluster formation pathways can be especially significant in cold and polluted areas, which can not only clarify the different roles of TFA in SA-DMA-based nucleation under broad atmospheric conditions, but also can reveal the potential implications of the usages of

[Figure]

Freon alternatives on NPF under a wide range of atmospheric conditions. In general, the manuscript is well written and is of broad interest to the readership of Atmospheric Chemistry and Physics. I can recommend publication in Atmospheric Chemistry and Physics after the following comments have been addressed.

Specific Comments: Lines 22-24: "Although sulfuric acid (SA) – dimethylamine (DMA) nucleation mechanism has been identified in the urban city of Shanghai, China (Yao et al., 2018), the nucleation mechanisms are not fully understood and species contributing to NPF under different environments remain to be studied" Please further elaborate why the nucleation mechanisms are still not fully understood even though the SA-DMA nucleation mechanism has been identified in Shanghai. Lines 29-30: "..., PFCAs are generally believed to be an important class of environmental contaminants present in various environments" Please provide relevant references for the atmospheric importance of PFCAs. Section 3.1: From Figure 1, it seems that the (SA)$_3$·(DMA)$_4$·(TFA)$_1$ clusters can be set as boundary clusters. Are there other clusters that can be set as boundary? The detailed boundary clusters that can grow out of the simulated system by Atmospheric Cluster Dynamics Code and the reasons should be illustrated. Line 137: The corresponding temperature and DMA concentration for the 13 times enhancement by TFA should be presented. Line 195 and line 196: The detailed growing way of clusters in SA-DMA-TFA cluster formation pathway should be further illustrated. The reasons for that some clusters involving TFA does not present in the main cluster formation pathway should be elaborated. Supplement, Table S5: Are these simulated results based on the thermodynamic parameters, such as Gibbs free formation energies ($\Delta$G), at the corresponding temperatures shown in Table S5? If so, the $\Delta$G of studied clusters at different temperatures of the studied cities in different months should be presented in the Supplement.

Technical corrections: Line 73: "The collision rate coefficients $\beta$i,j between clusters i and j were ..." should be "The collision rate coefficients, $\beta$i,j, between clusters i and j were ..." Line 83: "where Pref is the reference pressure (in this case 1 atm) where

the formation free energies. . ." should be "where Pref is the reference pressure (in this case 1 atm), at which the formation free energies. . ." Line 86: "3.1 In Influence of temperature and nucleation precursor concentrations on cluster stability and growth trend . . ." should be "3.1 In Influence of temperature and nucleation precursor concentrations on the stability and growth trend . . ." Line 135: ". . . temperature in spring and winter is relatively lower than other time all the year-round, respectively." should be ". . . temperatures in spring and winter are all relatively lower than other time all the year-round." Supplement, Line 1 and Line 3: The "ΔG" should be in italic, such as "ΔG".
* * *

---

## Referee Comment (RC2) · Anonymous Referee #2 · 5 Jan 2021

The Study of Liu et al. report the influence of atmospheric conditions on the role of a perfluorocarboxylic acid, namely trifluoroacetic acid (TFA) in sulfate-based aerosol formation. This study readily complements a previous study from the same authors (Lu et al. 2020) by extending the local atmospheric conditions to worldwide atmospheric conditions. They used density functional theory and dynamics simulation to show that the particle formation rate and the contributions of sulfuric acid – dimethylamine – TFA clusters to the cluster formation pathways can be effectively enhanced, especially in cold and mildly polluted regions. This study highlights the influence of the use of Freon alternatives on air quality and climate, and is of great interest for researchers in various fields within atmospheric and aerosol sciences. The study is well conducted and the

approach is sound. I recommend publication in Atmospheric Chemistry and Physics after the comments below have been addressed.

The referee comments are provided as a supplement.

Please also note the supplement to this comment:
https://acp.copernicus.org/preprints/acp-2020-1186/acp-2020-1186-RC2-supplement.pdf

**Supplement:**

**Referee Comments**

The Study of Liu et al. report the influence of atmospheric conditions on the role of a perfluorocarboxylic acid, namely trifluoroacetic acid (TFA) in sulfate-based aerosol formation. This study readily complements a previous study from the same authors (Lu et al. 2020) by extending the local atmospheric conditions to worldwide atmospheric conditions. They used density functional theory and dynamics simulation to show that the particle formation rate and the contributions of sulfuric acid – dimethylamine – TFA clusters to the cluster formation pathways can be effectively enhanced, especially in cold and mildly polluted regions. This study highlights the influence of the use of Freon alternatives on air quality and climate, and is of great interest for researchers in various fields within atmospheric and aerosol sciences. The study is well conducted and the approach is sound. I recommend publication in Atmospheric Chemistry and Physics after the comments below have been addressed.

**General comments**

It is well known from previous studies that plain SA-DMA clusters are insensitive to humidity. However, have the authors thought whether this situation may change if TFA is present in the cluster?

**Specific comments**

- Line 22: Include much earlier references including references from the same author and his group.
- Lines 22-23: SA-DMA nucleation has been observed in various places around the World. Why do the authors choose to constrain this fact solely to the urban city, Shanghai, China?
- Lines 24-25: A number of studies have identified earlier that some species (including methane sulfonic acid, sulfamic acid and glyoxylic acid) may enhance SA-based particle formation as well. You could list some of these studies to further put your study aim into context.

**Technical corrections**

- Line 28: "Due to the worldwide…" should be "Due to their worldwide…"
- Line 40: "Whereas TFA…" should be "However, TFA…"
- Line 43: "…with distances to the corresponding…" should be "…with distances to corresponding…"
- Lines 76: the publication year should be added to the reference Lu and Chen. No need to cite that same reference at line 78.
- Line 91: "It shows that the $\triangle G$ decreases…" should be "It shows that $\triangle G$ decreases…"
- Line 93: "…a cluster with monomer molecule…" should be "…a cluster with monomer molecule…"
- Lines 100-102: "The reason for this is that the influence of temperature variation on the evaporation coefficients ($\gamma$, where the temperature dependence is exponential (McGrath et al., 2012) is much greater than that on collision coefficients ($\beta$, where the dependence is in the square root of the temperature (McGrath et al., 2012))." should be "The reason for this is that

the influence of temperature variation on the evaporation coefficients ($\gamma$, where the temperature is in the exponential term (Eq. (3))) is much greater than that on collision coefficients ($\beta$, where the temperature is in the square root term (Eq. (2)))."

-Line 131: "…most of the year." should be "…most parts of the year."

-Line 132: "…This can be attributed to that the …" should be "…This can be attributed to the fact that the …"

-Line 135: "This can be attributed to that temperature in spring and winter is relatively lower than other time all the year-round, respectively." should be "This can be attributed to the fact that the temperature in spring and in winter is relatively lower than in other seasons all the year-round."

-Line 140: "…at the common temperatures…" should be "…at common temperatures…"

-Line 178: Unit should be $cm^{-3} s^{-1}$.

-Line 184-185: "…in the polluted atmosphere are shown in Figs. S1 and S2 in the Supplement, respectively." should be "…in the polluted atmosphere are respectively shown in Figs. S1 and S2 in the Supplement."

-Line 195: "…at the moderate value of…" should be "…at moderate values of…"

---

## Author Comment (AC1) · 2 Mar 2021

**Responses to Referee #1's comments**

We are grateful to the reviewers for their valuable and helpful comments on our manuscript "**Influence of atmospheric conditions on the role of trifluoroacetic acid in atmospheric sulfuric acid-dimethylamine nucleation**" (MS No.: acp-2020-1186). We have revised the manuscript carefully according to reviewers' comments. The point-to-point responses to the Referee #1's comments are summarized below:

**General Comments:**

Liu and co-workers investigated the influence of different atmospheric conditions on the role of trifluoroacetic acid (TFA) in sulfuric acid (SA) - dimethylamine (DMA) nucleation process using the Density Functional Theory combined with the Atmospheric Cluster Dynamics Code, which is based the previous study on that TFA can participate in SA-DMA-based nucleation under the local atmospheric temperature and nucleation precursor concentrations of Shanghai, China. This study reports the enhancement of particle formation rate by TFA and the contributions of SA-DMA-TFA cluster to the cluster formation pathways can be especially significant in cold and polluted areas, which can not only clarify the different roles of TFA in SA-DMA-based nucleation under broad atmospheric conditions, but also can reveal the potential implications of the usages of Freon alternatives on NPF under a wide range of atmospheric conditions. In general, the manuscript is well written and is of broad interest to the readership of Atmospheric Chemistry and Physics. I can recommend publication in Atmospheric Chemistry and Physics after the following comments have been addressed.

**Response:** We would like to thank the reviewer for the positive and valuable comments, and we have revised our manuscript accordingly.
* * *
**Specific Comments:**

**Comment 1.**

**Lines 22-24:** "Although sulfuric acid (SA) - dimethylamine (DMA) nucleation mechanism has been identified in the urban city of Shanghai, China (Yao et al., 2018), the nucleation mechanisms are not fully understood and species contributing to NPF under different environments remain to be studied" Please further elaborate why the nucleation mechanisms are still not fully understood even though the SA-DMA nucleation mechanism has been identified in Shanghai.

**Response:** According to the reviewer's suggestion, we now state in Lines 22-25 of the revised manuscript that "Although sulfuric acid (SA) - dimethylamine (DMA) nucleation mechanism has been observed in various places around the world (Yao et al., 2018; Deng et al., 2020; Brean et al., 2020), there were still a lot of species observed in the atmosphere but not be fully assigned molecular formulas because of their chemical complexity.", and in Lines 27-28 of the revised manuscript that "However, other possible species that may potentially enhance the NPF rates and the

corresponding nucleation mechanism still should be further explored."
* * *
**Comment 2.**

**Lines 29-30:** "…, PFCAs are generally believed to be an important class of environmental contaminants present in various environments" Please provide relevant references for the atmospheric importance of PFCAs.

**Response:** According to the reviewer's suggestion, the relevant references for the atmospheric importance of PFCAs has been added in Lines 30-31.
* * *
**Comment 3.**

**Section 3.1:** From Figure 1, it seems that the $(SA)_3 \cdot (DMA)_4 \cdot (TFA)_1$ clusters can be set as boundary clusters. Are there other clusters that can be set as boundary? The detailed boundary clusters that can grow out of the simulated system by Atmospheric Cluster Dynamics Code and the reasons should be illustrated.

**Response:** We now state in Lines 113-116 of the revised manuscript that "The boundary of ACDC simulation should be set as the smallest clusters that are stable enough to grow outside of the simulated system (McGrath et al., 2012). Based on the stability of the studied clusters, $(SA)_4 \cdot (DMA)_3$ and $(SA)_3 \cdot (DMA)_4 \cdot (TFA)_1$ clusters are set to be boundary, which is consistent with that in the previous study (Lu et al., 2020)."
* * *
**Comment 4.**

**Line 137:** The corresponding temperature and DMA concentration for the 13 times enhancement by TFA should be presented.

**Response:** The corresponding temperature and DMA concentration for the 13 times enhancement by TFA have been added in Lines 148-149 of the revised manuscript that "… more than 13 times in January at [DMA] of $4.69 \times 10^8$ molecules cm$^{-3}$ and 265 K."
* * *
**Comment 5.**

**Line 195 and line 196:** The detailed growing way of clusters in SA-DMA-TFA cluster formation pathway should be further illustrated. The reasons for that some clusters involving TFA does not present in the main cluster formation pathway should be elaborated.

**Response:** We now state in Lines 209-212 of the revised manuscript that "The clusters involving TFA can be formed initially by the addition of one TFA molecule to $(SA)_1 \cdot (DMA)_1$ cluster, and continue growing by the addition of one DMA molecule and then the addition of one $(SA)_1 \cdot (DMA)_1$ cluster. Other studied clusters involving TFA can't present in the main cluster formation pathway because of relatively low stability."
* * *
**Comment 6.**

**Supplement, Table S5:** Are these simulated results based on the thermodynamic parameters, such as Gibbs free formation energies ($\Delta G$), at the corresponding

temperatures shown in Table S5? If so, the $\Delta G$ of studied clusters at different temperatures of the studied cities in different months should be presented in the Supplement.

**Response:** We now list the Gibbs free formation energies ($\Delta G$) in Table S10 of the revised Supplement at the relevant temperatures of the studied cities in different months shown in Table S9.
* * *
**Technical corrections:**

**Comment 7.**

**Line 73:** "The collision rate coefficients $\beta_{i,j}$ between clusters $i$ and $j$ were ..." should be "The collision rate coefficients, $\beta_{i,j}$, between clusters $i$ and $j$ were ..."

**Response:** "The collision rate coefficients $\beta_{i,j}$ between clusters $i$ and $j$ were ..." has been corrected as "The collision rate coefficients, $\beta_{i,j}$, between clusters $i$ and $j$ were ..." in Line 76 of the revised manuscript.
* * *
**Comment 8.**

**Line 83:** "where $P_{ref}$ is the reference pressure (in this case 1 atm) where the formation free energies..." should be "where $P_{ref}$ is the reference pressure (in this case 1 atm), at which the formation free energies..."

**Response:** "where $P_{ref}$ is the reference pressure (in this case 1 atm) where the formation free energies..." has been corrected as "where $P_{ref}$ is the reference pressure (in this case 1 atm), at which the formation free energies..." in Line 85 of the revised manuscript.
* * *
**Comment 9.**

**Line 86:** "3.1 In Influence of temperature and nucleation precursor concentrations on cluster stability and growth trend ..." should be "3.1 In Influence of temperature and nucleation precursor concentrations on the stability and growth trend ..."

**Response:** According to the reviewer's suggestion and based on the addition of the discussion on influence of humidity in Section 3.1, "3.1 In Influence of temperature and nucleation precursor concentrations on cluster stability and growth trend ..." has been corrected as "3.1 Influence of atmospheric conditions on the stability and growth trend ..." in Line 88 of the revised manuscript.
* * *
**Comment 10.**

**Line 135:** "… temperature in spring and winter is relatively lower than other time all the year-round, respectively." should be "… temperatures in spring and winter are all relatively lower than other time all the year-round."

**Response:** "… temperature in spring and winter is relatively lower than other time all the year-round, respectively." has been corrected as "… temperature in spring and in winter is relatively lower than in other seasons all the year round." in Lines 146-147 of the revised manuscript.
* * *
**Comment 11.**

**Supplement, Line 1 and Line 3:** The "ΔG" should be in italic, such as "$\Delta G$".

**Response:** The "ΔG" has been corrected in italic as "$\Delta G$" in Line 29 and Line 31 of the revised Supplement.

---

## Author Comment (AC2) · 2 Mar 2021

**Responses to Referee #2's comments**

We are grateful to the reviewers for their valuable and helpful comments on our manuscript "**Influence of atmospheric conditions on the role of trifluoroacetic acid in atmospheric sulfuric acid-dimethylamine nucleation**" (MS No.: acp-2020-1186). We have revised the manuscript carefully according to reviewers' comments. The point-to-point responses to the Referee #2's comments are summarized below:

**Referee Comments:**
The Study of Liu et al. report the influence of atmospheric conditions on the role of a perfluorocarboxylic acid, namely trifluoroacetic acid (TFA) in sulfate-based aerosol formation. This study readily complements a previous study from the same authors (Lu et al. 2020) by extending the local atmospheric conditions to worldwide atmospheric conditions. They used density functional theory and dynamics simulation to show that the particle formation rate and the contributions of sulfuric acid – dimethylamine – TFA clusters to the cluster formation pathways can be effectively enhanced, especially in cold and mildly polluted regions. This study highlights the influence of the use of Freon alternatives on air quality and climate, and is of great interest for researchers in various fields within atmospheric and aerosol sciences. The study is well conducted and the approach is sound. I recommend publication in Atmospheric Chemistry and Physics after the comments below have been addressed.
**Response:** We would like to thank the reviewer for the positive and valuable comments, and we have revised our manuscript accordingly.
* * *
**General comments**
**Comment 1.**
It is well known from previous studies that plain SA-DMA clusters are insensitive to humidity. However, have the authors thought whether this situation may change if TFA is present in the cluster?
**Response:** According to the reviewer's suggestion, we now state in Lines 107-113 of the revised manuscript that "In addition, in order to evaluate the influence of relative humidity on the formation kinetics of SA-DMA-TFA system, the hydrated key clusters have also been studied. The relative hydrate distributions, the ratios of effective collision and evaporation coefficients weighted average over the hydrate distributions (Paasonen et al., 2012) and Cartesian coordinates of hydrated clusters are listed in Tables S5-S6, Table S7 and Table S8 in the Supplement, respectively. From the relevant detailed discussion shown in Section S1 in the Supplement, we can see that the stability of the studied SA-DMA-TFA clusters is insensitive to relative humidity, which is similar to the SA-DMA clusters (Olenius et al., 2017). Thus, only the unhydrated SA-DMA-TFA clusters are considered in the following discussion."

Detailed analysis about the influence of humidity on the studied SA-DMA-TFA clusters has been shown in Section 1 of the revised Supplement that "From the ratios

of $\beta \cdot C / \Sigma \gamma$ for the studied SA-DMA-TFA clusters (Fig. 1), it can be seen that the $(SA)_2 \cdot (DMA)_3 \cdot (TFA)_1$ and $(SA)_1 \cdot (DMA)_2 \cdot (TFA)_1$ clusters are relatively stable against evaporation and can be able to grow into larger clusters. The $(SA)_1 \cdot (DMA)_1 \cdot (TFA)_1$ cluster is the initial and key cluster to form $(SA)_2 \cdot (DMA)_3 \cdot (TFA)_1$ and $(SA)_1 \cdot (DMA)_2 \cdot (TFA)_1$ clusters as shown in Fig. 4, and this cluster formation pathway through $(SA)_1 \cdot (DMA)_1 \cdot (TFA)_1$ involves a modest thermodynamic barrier shown by the previous study (Lu et al. 2020). Hence, the formation of $(SA)_1 \cdot (DMA)_1 \cdot (TFA)_1$ cluster is the limiting step in SA-DMA-TFA new particle formation, which is similar to the analysis on the limiting step of $(SA)_n \cdot (Base)_n$ system (Elm, 2017). Therefore, in order to understand the influence of relative humidity (RH) on the SA-DMA-TFA system, the evaluation the influence of RH on the formation kinetics of $(SA)_1 \cdot (DMA)_1 \cdot (TFA)_1$ cluster is of significance. Herein, the kinetic property of hydrated clusters relevant to the formation of the hydrated $(SA)_1 \cdot (DMA)_1 \cdot (TFA)_1$ clusters, which involves collisions of smaller clusters with monomers and evaporation of monomer from the larger cluster, has been studied. The relative hydrate distributions of the clusters (Tables S5-S6) and the effective collision and evaporation coefficients (Table S7) as the weighted average over the hydrate distributions (Paasonen et al., 2012) are calculated at 280 K and 260 K. The Cartesian coordinates of these studied hydrated clusters are listed in Table S8. The relative distributions of the unhydrated $(SA)_1 \cdot (DMA)_1 \cdot (TFA)_1$ and $(SA)_1 \cdot (DMA)_2 \cdot (TFA)_1$ clusters are more than 50%, which are higher than those of the corresponding hydrated clusters at different RHs (20%, 40%, 60%, 80%, and 100%) and different temperatures (280 K and 260 K). Hence, most of $(SA)_1 \cdot (DMA)_1 \cdot (TFA)_1$ and $(SA)_1 \cdot (DMA)_2 \cdot (TFA)_1$ clusters are unhydrated in the atmosphere. Furthermore, the ratios of effective collision frequencies with nucleation monomers versus total effective evaporation frequencies ($\beta_{eff} C / \Sigma \gamma_{eff}$) of these two key clusters almost vary slightly within one order of magnitude at different RHs and different temperatures (Table S7). Therefore, the studied SA-DMA-TFA system is insensitive to the variation of humidity, which is similar to the SA-DMA system (Olenius et al., 2017)."
* * *
**Specific comments**

**Comment 2.**

**- Line 22:** Include much earlier references including references from the same author and his group.

**Response:** According to the reviewer's suggestion, the relevant earlier references including references from the same author and his group have been added in Line 22 of the revised manuscript.
* * *
**Comment 3.**

**- Lines 22-23:** SA-DMA nucleation has been observed in various places around the World. Why do the authors choose to constrain this fact solely to the urban city, Shanghai, China?

**Response:** SA-DMA nucleation has indeed been observed in various places around the World. Thus, according to the reviewer's suggestion, we have rephrased this

sentence in Lines 22-25 of the revised manuscript that "Although sulfuric acid (SA) - dimethylamine (DMA) nucleation mechanism has been observed in various places around the world (Yao et al., 2018; Deng et al., 2020; Brean et al., 2020), there were still a lot of species observed in the atmosphere but not be fully assigned molecular formulas because of their chemical complexity."
* * *
**Comment 4.**

**-Lines 24-25:** A number of studies have identified earlier that some species (including methane sulfonic acid, sulfamic acid and glyoxylic acid) may enhance SA-based particle formation as well. You could list some of these studies to further put your study aim into context.

**Response:** We now state in Lines 25-28 of the revised manuscript that "Previous studies have identified that some acids, such as methane sulfonic acid, sulfamic acid and glyoxylic acid, can enhance the SA-based particle formation (Bork et al., 2014; Li et al., 2018; Liu et al., 2017). However, other possible species that may potentially enhance the NPF rates and the corresponding nucleation mechanism still should be further explored."
* * *
**Technical corrections**

**Comment 5.**

**- Line 28:** "Due to the worldwide…" should be "Due to their worldwide…"

**Response:** "Due to the worldwide…" has been corrected as "Due to their worldwide…" in Line 31 of the revised manuscript.
* * *
**Comment 6.**

**- Line 40:** "Whereas TFA…" should be "However, TFA…"

**Response:** "Whereas TFA…" has been corrected as "However, TFA…" in Line 43 of the revised manuscript.
* * *
**Comment 7.**

**-Line 43:** "…with distances to the corresponding…" should be "…with distances to corresponding…"

**Response:** "…with distances to the corresponding…" has been corrected as "…with distances to corresponding…" in Line 46 of the revised manuscript.
* * *
**Comment 8.**

**- Lines 76:** the publication year should be added to the reference Lu and Chen. No need to cite that same reference at line 78.

**Response:** The publication year has been added to the reference Lu and Chen in Line 79 of the revised manuscript. The same reference in Line 81 has been deleted in the revised manuscript.
* * *
**Comment 9.**

**- Line 91:** "It shows that the $\Delta G$ decreases…" should be "It shows that $\Delta G$

decreases…"

**Response:** "It shows that the ΔG decreases…" has been corrected as "It shows that $\Delta G$ decreases…" in Line 92 of the revised manuscript.
* * *
**Comment 10.**
**- Line 93:** "…a cluster with monomer molecule…" should be "…a cluster with monomer molecule…"

**Response:** "…a cluster with monomer molecule…" has been corrected as "…a cluster with monomer molecule…" in Line 94 of the revised manuscript.
* * *
**Comment 11.**
**-Lines 100-102:** "The reason for this is that the influence of temperature variation on the evaporation coefficients ($\gamma$, where the temperature dependence is exponential (McGrath et al., 2012) is much greater than that on collision coefficients ($\beta$, where the dependence is in the square root of the temperature (McGrath et al., 2012))." should be "The reason for this is that the influence of temperature variation on the evaporation coefficients ($\gamma$, where the temperature is in the exponential term (Eq. (3))) is much greater than that on collision coefficients ($\beta$, where the temperature is in the square root term (Eq. (2)))."

**Response:** According to the reviewer's suggestion, "The reason for this is that the influence of temperature variation on the evaporation coefficients ($\gamma$, where the temperature dependence is exponential (McGrath et al., 2012) is much greater than that on collision coefficients ($\beta$, where the dependence is in the square root of the temperature (McGrath et al., 2012))." has been corrected as "The reason for this is that the influence of temperature variation on the evaporation coefficients ($\gamma$, where the temperature is in the exponential term (Eq. (3))) is much greater than that on collision coefficients ($\beta$, where the temperature is in the square root term (Eq. (2))) (McGrath et al., 2012)." in Lines 101-103 of the revised manuscript.
* * *
**Comment 12.**
**-Line 131:** "…most of the year." should be "…most parts of the year."

**Response:** "…most of the year." has been corrected as "…most parts of the year." in Line 142 of the revised manuscript.
* * *
**Comment 13.**
**-Line 132:** "…This can be attributed to that the …" should be "…This can be attributed to the fact that the …"

**Response:** "…This can be attributed to that the …" has been corrected as "…This can be attributed to the fact that the …" in Line 143 of the revised manuscript.
* * *
**Comment 14.**
**-Line 135:** "This can be attributed to that temperature in spring and winter is relatively lower than other time all the year-round, respectively." should be "This can be attributed to the fact that the temperature in spring and in winter is relatively lower

than in other seasons all the year round."

**Response:** "This can be attributed to that temperature in spring and winter is relatively lower than other time all the year-round, respectively." has been corrected as "This can be attributed to the fact that the temperature in spring and in winter is relatively lower than in other seasons all the year round." in Lines 146-147 of the revised manuscript.
* * *
**Comment 15.**

**-Line 140:** "…at the common temperatures…" should be "…at common temperatures…"

**Response:** "…at the common temperatures…" has been corrected as "…at common temperatures…" in Line 152 of the revised manuscript.
* * *
**Comment 16.**

**-Line 178:** Unit should be $cm^{-3} s^{-1}$.

**Response:** Unit has been corrected as "$cm^{-3} s^{-1}$" in Line 190 of the revised manuscript.
* * *
**Comment 17.**

**-Line 184-185:** "…in the polluted atmosphere are shown in Figs. S1 and S2 in the Supplement, respectively." should be "…in the polluted atmosphere are respectively shown in Figs. S1 and S2 in the Supplement."

**Response:** "…in the polluted atmosphere are shown in Figs. S1 and S2 in the Supplement, respectively." has been corrected as "…in the polluted atmosphere are respectively shown in Figs. S1 and S2 in the Supplement." in Lines 196-197 of the revised manuscript.
* * *
**Comment 18.**

**-Line 195:** "…at the moderate value of…" should be "…at moderate values of…"

**Response:** "…at the moderate value of…" has been corrected as "…at moderate values of…" in Lines 207 of the revised manuscript.

---

## Author Response (AR2)

Dear Editor,

We thank you for taking the time to review our manuscript "**Influence of atmospheric conditions on the role of trifluoroacetic acid in atmospheric sulfuric acid-dimethylamine nucleation**" (MS No.: acp-2020-1186). We have carefully checked out the language of the manuscript and corrected some typos and grammatical issue in the revised manuscript accordingly. The point-by-point revisions are summarized below in blue text.

**Comments to the Author:**

The authors have addressed all the scientific issues raised by the referees in a sufficient detail. However, the text (especially the new one but also the original text) still contains some typos or minor grammatical issue. Therefore, the authors should carefully check out the language of the paper once more. After that, the paper will be acceptable for publication.

Editor

**Responses:**

**Line 9:** "The enhancement of particle formation rate by TFA …" has been corrected as "The enhancement on particle formation rate by TFA …" in Line 9 of the revised manuscript.
* * *
**Line 10:** "… can be up to as much as 227 times …" has been corrected as "… can be up to 227 times …" in Line 10 of the revised manuscript.
* * *
**Lines 11-12:** "… such as in winter or at relatively high atmospheric boundary layer and in megacities …" has been corrected as "… such as in winter, at the relatively high atmospheric boundary layer, or in megacities …" in Lines 11-12 of the revised manuscript.
* * *
**Line 24:** "… there were still a lot of species observed in the atmosphere but not be fully assigned …" has been corrected as "… there are still a lot of species observed in the atmosphere but not fully assigned …"in Line 24 of the revised manuscript.
* * *
**Line 27:** "… other possible species that may potentially enhance …" has been corrected as "… other species that can potentially enhance …" in Line 27 of the revised manuscript.
* * *
**Line 33:** "… envirnments …" has been corrected as "… environments …" in Line 33 of the revised manuscript.
* * *
**Line 42:** "… the role of TFA in SA-DMA-based NPF process …" has been corrected

as "… the role of TFA in the SA-DMA-based NPF process …" in Line 42 of the revised manuscript.
* * *
**Lines 45-46:** "Temperature can change with the variations of seasons, altitudes and climates, and nucleation precursor concentrations can vary with distances to corresponding sources and altitudes." has been corrected as "Not only can temperature change with the variations of seasons, altitudes, and climates, but also nucleation precursor concentrations can vary with altitudes and distances to corresponding sources." in Lines 45-46 of the revised manuscript.
* * *
**Line 49:** "… power plant or industry, but it is relatively low in the areas …" has been corrected as "… power plants or industries, but it is relatively low in areas …" in Line 49 of the revised manuscript.
* * *
**Line 55:** "… $(Acid)_m \cdot (Base)_n$, where $0 \leq n \leq m \leq 3$…" has been corrected as "… $(Acid)_m \cdot (Base)_n$ $(0 \leq n \leq m \leq 3)$ …" in Line 55 of the revised manuscript.
* * *
**Line 59:** "… estimate thermochemistry…" has been corrected as "… estimate the thermochemistry …" in Line 59 of the revised manuscript.
* * *
**Line 62:** "… were corrected using RI-CC2 method and aug-cc-pV(T+d)Z basis …" has been corrected as "… were corrected using the RI-CC2 method and the aug-cc-pV(T+d)Z basis …" in Line 62 of the revised manuscript.
* * *
**Line 64:** "… at RI-CC2/aug-cc-pV(T+d)Z level of theory …" has been corrected as "… at the RI-CC2/aug-cc-pV(T+d)Z level of theory…" in Line 64 of the revised manuscript.
* * *
**Line 65:** "… at 280 K are taken from …" has been corrected as "… at 280 K were taken from …" in Lines 65-66 of the revised manuscript.
* * *
**Line 69:** "… by integrating numerically the birth-death equation …" has been corrected as "… by integrating the birth-death equation numerically …" in Line 69 of the revised manuscript.
* * *
**Line 73:** "… $\gamma_{i \to j}$ is the $\gamma$ evaporation …" has been corrected as "… $\gamma_{i \to j}$ is the evaporation …" in Line 73 of the revised manuscript.
* * *
**Line 80:** "… the temperature and $\mu = mimj = (mi + mj)$ …" has been corrected as "… the temperature, and $\mu = mimj = (mi + mj)$ …" in Line 80 of the revised manuscript.
* * *
**Line 81:** "… most distant atoms in cluster …" has been corrected as "… most distant atoms in the cluster …" in Line 81 of the revised manuscript.
* * *
**Line 94:** "… a cluster with monomer molecule …" has been corrected as "… a cluster with a monomer molecule …" in Line 94 of the revised manuscript.
* * *
**Line 99:** "… temperatures as an example shown in Fig. 1 to study …" has been corrected as "… temperatures shown in Fig. 1 as an example to study …" in Line 99 of the revised manuscript.
* * *
**Lines 107-108:** "… formation kinetics of SA-DMA-TFA …" has been corrected as "… formation kinetics of the SA-DMA-TFA …" in Line 107 of the revised manuscript.
* * *
**Line 109:** "… effective collision and evaporation coefficients weighted average over the hydrate distributions (Paasonen et al., 2012) and …" has been corrected as "… effective collision coefficients to evaporation coefficients weighted average over the hydrate distributions (Paasonen et al., 2012), and …" in Lines 108-109 of the revised manuscript.
* * *
**Line 113:** "The boundary of ACDC simulaion …" has been corrected as "The boundary of the ACDC simulation …" in Line 113 of the revised manuscript.
* * *
**Line 115:** "… clusters are set to be boundary …" has been corrected as "clusters are set to be the boundary …" in Line 115 of the revised manuscript.
* * *
**Lines 121-123:** "… rates of SA-DMA-TFA system have been investigated, as described below." has been corrected as "… rates of the SA-DMA-TFA system have been investigated below." in Lines 121-123 of the revised manuscript.
* * *
**Lines 127-128:** "The lifetime of TFA and SA are relatively long enough to ensure their atmospheric concentration to be in the present studied concentration range." has been corrected as "The lifetime of TFA and SA is long enough to keep their atmospheric concentrations within the studied range." in Line 127 of the revised manuscript.
* * *
**Line 133:** "… have been simulated at the [DMA] of 5 times the corresponding predicted [DMA] ..." has been corrected as "… have been simulated at five times the corresponding predicted [DMA] ..." in Lines 132-133 of the revised manuscript.
* * *
**Line 135:** "… as an example (Fig. 2) ..." has been corrected as "… as examples (Fig. 2). ..." in Line 134 of the revised manuscript.
* * *
**Line 137:** "The [SA] and [TFA] were chosen to be relatively low …" has been corrected as "The SA concentration ([SA]) and TFA concentration ([TFA]) were chosen to be relatively low ..." in Lines 136-137 of the revised manuscript.
* * *
**Lines 142-143:** "… and enhancement on particle formation rate by TFA of Beijing are almost all higher than those of the other three studied cities …" has been corrected as "… and the enhancement on particle formation rate by TFA in Beijing are almost all higher than those in the other three studied cities ..." in Lines 142-143 of the revised manuscript.
* * *
**Line 144:** "… [DMA] and temperature in Beijing is relatively higher and lower than those of the other three studied cities…" has been corrected as "… [DMA] and temperature in Beijing are relatively higher and lower than those in the other three studied cities ..." in Line 144 of the revised manuscript.
* * *
**Lines 146-147:** "This can be attributed to the fact that the temperature in spring and in winter is relatively lower than in other seasons all the year round." has been corrected as "This can be attributed to the fact that the temperature in spring and winter is relatively lower than that in other seasons all the year round." in Lines 146-147 of the revised manuscript.
* * *
**Lines 156-157:** "… The common atmospheric concentrations of DMA monomer ([DMA]) are in the range of …" has been corrected as "… The common [DMA] in the atmosphere is in the range of ..." in Lines 156-157 of the revised manuscript.
* * *
**Line 159:** "…atmospheric concentrations of SA ([SA]) and TFA ([TFA])." has been corrected as "… atmospheric [SA]) and [TFA]." in Line 159 of the revised manuscript.
* * *
**Line 160:** "… the enhancement of particle formation rate by TFA …" has been corrected as "… the enhancement on particle formation rate by TFA ..." in Line 160 of the revised manuscript.
* * *
**Lines 168-169:** "… its main sources is the coal-fired power plant and the industry …" has been corrected as "… its main sources are the coal-fired power plants and the industries ..." in Lines 168-169 of the revised manuscript.
* * *
**Line 173:** "… negative dependencies of particle formation rate …" has been corrected as "… negative dependencies of the particle formation rate ..." in Line 173 of the revised manuscript.
* * *
**Line 181:** "… precursors except SA …" has been corrected as "… precursors besides SA ..." in Line 181 of the revised manuscript.
* * *
**Line 190:** "… more than 17 $cm^{-3}\,s^{-1}$ and the corresponding enhancement …" has been corrected as "… more than 17 $cm^{-3}\,s^{-1}$, and the corresponding enhancement ..." in Line 190 of the revised manuscript.
* * *
**Line 198:** "… nucleation precursor concentrations on them are the same." has been corrected as "… nucleation precursor concentrations on them at different CSs are the same." in Lines 198-199 of the revised manuscript.
* * *
**Line 211-212:** "… because of relatively low stability." has been corrected as "… because of their relatively low stability." in Lines 212-213 of the revised manuscript.
* * *
**Line 245:** "… the increase of [DMA] and [TFA], and decrease of [SA]." has been corrected as "… the increase of [DMA] and [TFA], and the decrease of [SA]." in Lines 246-247 of the revised manuscript.
* * *
**Line 255:** "… inspires on the potential underlying influence …" has been corrected as "… inspires the potential influence …" in Line 256 of the revised manuscript.
* * *
**Supplement, Line 9:** "… the analysis about the limiting step …" has been corrected as "… the analysis on the limiting step …" in Line 9 of the revised supplement.
* * *
**Supplement, Line 14:** "… evaporation of monomer from the larger cluster …" has been corrected as "… evaporation of monomers from the larger clusters …" in Line 14 of the revised supplement.
* * *
**Supplement, Line 15-16:** "… effective collision and evaporation coefficients (Table S7) as the weighted average over …" has been corrected as "… effective collision coefficients and evaporation coefficients (Table S7) weighted average over …" in Lines 15-16 of the revised supplement.
* * *
**Supplement, Lines 29-30:** The information of the theoretical method has been added as "… the studied clusters at different temperatures at the RI-CC2/aug-cc-pV(T + d)Z//M06-2X/6-311++G(3df,3pd) level of theory …" in Lines 29-30 of the revised supplement.
* * *
**Supplement, Line 57:** The information of the theoretical method has been added as "… Cartesian coordinates of the most stable hydrated clusters in the present study at the M06-2X/6-311++G(3df,3pd) level of theory …" in Lines 57-58 of the revised supplement.
* * *
Sincerely,
Xiuhui Zhang
Key Laboratory of Cluster Science Ministry of Education of China
School of Chemistry and Chemical Engineering
Beijing Institute of Technology
Beijing 100081, P.R. China
Email: zhangxiuhui@bit.edu.cn